# AW-Opt: Learning Robotic Skills with Imitation and Reinforcement at Scale

**Yao Lu**[1], **Karol Hausman**[1], **Yevgen Chebotar**[1], **Mengyuan Yan**[2], **Eric Jang**[1], **Alexander Herzog**[2], **Ted Xiao**[1], **Alex Irpan**[1], **Mohi Khansari**[2], **Dmitry Kalashnikov**[1], **Sergey Levine**[1,3]

[1]Robotics at Google    [2]X, The Moonshot Factory    [3]UC Berkeley

**Abstract:** Robotic skills can be learned via imitation learning (IL) using user-provided demonstrations, or via reinforcement learning (RL) using large amounts of autonomously collected experience. Both methods have complementary strengths and weaknesses: RL can reach a high level of performance, but requires exploration, which can be very time consuming and unsafe; IL does not require exploration, but only learns skills that are as good as the provided demonstrations. Can a single method combine the strengths of both approaches? A number of prior methods have aimed to address this question, proposing a variety of techniques that integrate elements of IL and RL. However, scaling up such methods to complex robotic skills that integrate diverse offline data and generalize meaningfully to real-world scenarios still presents a major challenge. In this paper, our aim is to test the scalability of prior IL + RL algorithms and devise a system based on detailed empirical experimentation that combines existing components in the most effective and scalable way. To that end, we present a series of experiments aimed at understanding the implications of each design decision, so as to develop a combined approach that can utilize demonstrations and heterogeneous prior data to attain the best performance on a range of real-world and realistic simulated robotic problems. Our complete method, which we call AW-Opt, combines elements of advantage-weighted regression [1, 2] and QT-Opt [3], providing a unified approach for integrating demonstrations and offline data for robotic manipulation. Please see https://awopt.github.io for more details.

**Keywords:** Imitation Learning, Reinforcement Learning, Robot Learning

## 1    Introduction

Learning methods for robotic control are conventionally divided into two groups: methods based on autonomous trial-and-error (reinforcement learning), and methods based on imitating user-provided demonstrations (imitation learning). These two approaches have complementary strengths and weaknesses. Reinforcement Learning (RL) enables robots to improve autonomously, but introduce significant challenges with exploration and stable learning. Imitation learning (IL) methods provide for more stable learning from expert demonstrations, but cannot improve from failures. This could potentially make covering the full distribution of data difficult.

Prior works have sought to combine elements of IL and RL into a single algorithm (IL+RL), by pre-training RL policies with demonstrations [4], incorporating demonstration data into off-policy RL algorithms [5], and combining IL and RL loss functions into a single algorithm [6]. While such algorithms have been based on both policy gradients and value-based off-policy methods, the latter are particularly relevant in robotics settings due to their ability to leverage both demonstrations and the robot's own prior experience. A number of such methods have shown promising results in simulation and in the real world [7], but they often require extensive parameter tuning and, as we show in our experiments, are difficult to deploy at scale to solve multiple tasks in diverse scenarios.

In this paper, our goal is to develop, through extensive experimentation, a complete and scalable system for integrating IL and RL that enables learning robotic control policies from both demonstration data and suboptimal experience. Our aim is not to propose new principles for algorithms that combine IL and RL, but rather to determine how to best combine the components of existing methods to enable effective large-scale robotic learning. We begin our investigation with two exist-

5th Conference on Robot Learning (CoRL 2021), London, UK.

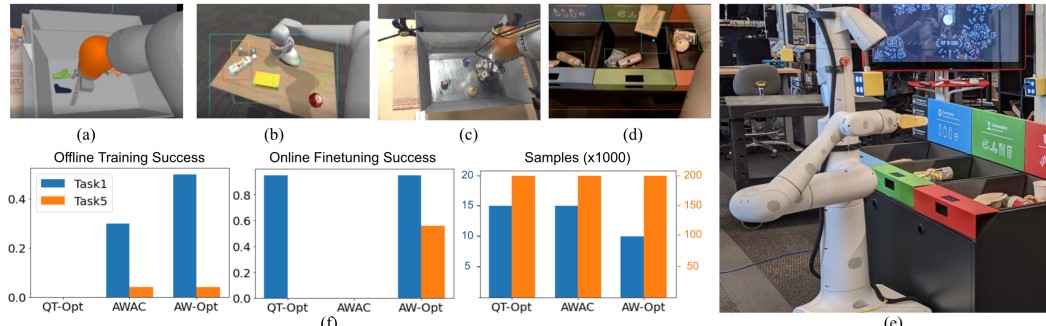

**Figure 1:** We study algorithms that combine imitation and reinforcement learning in the context of high-dimensional visual manipulation tasks. Existing RL algorithms such as QT-Opt [3] can solve simple tasks from scratch with a large amount of training data, but fail to take advantage of high-quality demonstrations data provided up-front and fail to solve difficult tasks. Offline RL methods that combine imitation learning and RL, such as Advantage-Weighted Actor-Critic (AWAC) [2], can learn good initial policies from demonstrations, but encounter difficulty when further fine-tuned with RL on large-scale robotic tasks, such as the diverse grasping scenarios in this figure. We present AW-Opt, an actor-critic algorithm that achieves stable learning in both offline imitation learning and RL fine-tuning phases. We evaluate our method on five different simulated (a,b) and real-world (c,d) visual grasping tasks. (a) Task 1: simulated indiscriminate grasping (b) Task 2: simulated instance grasping (c) Task 3: real indiscriminate grasping (d) Task 4 and Task 5: real instance grasping and grasping from a specific location (e) Running a real-world policy (f) Comparing success rates after initial offline imitation phase, final RL fine-tuning phase, and the number of samples needed to converge

ing methods: AWAC [2], which provides an integrated framework for combining IL and RL, and QT-Opt [3], which provides a scalable system for robotic reinforcement learning. We observe that neither method by itself provides a complete solution to large-scale RL with demonstrations. We then systematically study the decisions in these methods, identify their shortcomings, and develop a hybrid approach that we call AW-Opt. The individual design decisions that comprise AW-Opt may appear minor in isolation, but we show that combining these decisions significantly improves on the prior approaches, and scales effectively to learning high-capacity generalizable policies in multiple real and simulated environments shown in Fig. 1 (a-d).

The main contributions of this work consist of a detailed analysis of the design decisions that influence performance and scalability of IL+RL methods, leading to the development of AW-Opt, a hybrid algorithm that combines insights from multiple prior approaches to achieve scalable and effective robotic learning from demonstrations, suboptimal offline data, and online exploration. Not only can AW-Opt be successfully initialized from demonstrations (initial success in Fig. 1(f)), but also it responds well to RL fine-tuning (final success in Fig. 1(f)) while being sample-efficient.

In addition to the design of AW-Opt, we hope that the analysis, ablation experiments, and detailed evaluation of individual design decisions that we present in this work would help to guide the development of effective and scalable robotic learning algorithms. Our experimental results evaluate our method, as well as ablations, across a range of simulated and real-world domains, including two different real-world platforms and large datasets with hundreds of thousands of training episodes.

## 2   Related Work

Prior works have proposed a number of methods for learning robotic skills via imitation of experts [8, 9, 10, 11, 12, 13], and via reinforcement learning [14, 15, 16, 17, 18]. In the latter category, *off-policy* RL methods are especially appealing for large-scale robot learning, since they allow for data reuse with large, diverse datasets [3, 19, 20]. Initially, methods that combined imitation learning (IL) and reinforcement learning (RL) have tended to treat the two data sources differently, using IL demos only at initialization [4, 5, 21], or using different losses for IL demos vs. RL episodes [6]. Other methods have used a different approach: combining both expert demos and suboptimal prior data into a single dataset used for off-policy value-based learning [7]. Offline RL methods, such as CQL [22], QT-Opt [3], and AWR [1] often use this approach, along with constrained optimization and regularization to encourage staying close to the data distribution [23, 24, 25, 26, 27]. Our method builds on these approaches, due to their effectiveness, simplicity, and easier data management. Regardless of whether the data consists of optimal demonstrations, suboptimal human-provided demos, or very suboptimal robot experience, all data can be treated the same way.

Our goal is to scale up IL+RL methods to large datasets with many objects and varied environments. Standard RL methods have proven difficult to scale up to the same level as modern methods in computer vision [28, 29] and natural language processing [30, 31, 32], where datasets are diverse and open-world, and models have millions or billions of parameters. Taking RL to such settings requires large, distributed RL systems, which often require careful design decisions in their underlying RL methods. While standard methods such as actor-critic have been scaled up in this way [33, 34, 35, 36], we show in this work that scaling IL+RL algorithms requires a certain amount of care, especially in robotic settings where collecting real-world experience is costly. Naïvely incorporating demonstrations into an RL system (QT-Opt in our experiments) does not by itself yield good results. Similarly, utilizing prior IL+RL methods (such as AWAC [2] in our experiments) scales poorly. We provide a detailed empirical evaluation of various design decisions underlying IL+RL methods to develop a hybrid approach, which we call AW-Opt, that combines the scalability of QT-Opt with the ability to seamlessly incorporate suboptimal experience and bootstrap from demonstrations.

## 3 Preliminaries

Let $\mathcal{M} = (\mathcal{S}, \mathcal{A}, P, R, p_0, \gamma, T)$ define a Markov decision process (MDP), where $\mathcal{S}$ and $\mathcal{A}$ are state and action spaces, $P : \mathcal{S} \times \mathcal{A} \times \mathcal{S} \to \mathbb{R}_+$ is a state-transition probability function, $R : \mathcal{S} \times \mathcal{A} \to \mathbb{R}$ is a reward function, $p_0 : \mathcal{S} \to \mathbb{R}_+$ is an initial state distribution, $\gamma$ is a discount factor, and $T$ is the task horizon. We use $\tau = (s_0, a_0, \ldots, s_T, a_T)$ to denote a trajectory of states and actions and $R(\tau) = \sum_{t=0}^{T} \gamma^t R(s_t, a_t)$ to denote the trajectory reward. Reinforcement learning (RL) methods find a policy $\pi(a|s)$ that maximizes the expected discounted reward over trajectories induced by the policy: $\mathbb{E}_\pi[R(\tau)]$ where $s_0 \sim p_0, s_{t+1} \sim P(s_{t+1}|s_t, a_t)$ and $a_t \sim \pi(a_t|s_t)$.

In imitation learning, demonstrations are provided to facilitate learning an optimal policy, which can be produced through teleoperation [37, 38] or kinesthetic teaching [39, 12]. Given a data set of demonstration state-action tuples $(s, a) \sim \mathcal{D}_{demo}$, standard IL methods aim to directly approximate the expert's $\pi^*(a|s)$ via supervised learning. Another way of learning from prior data has been explored in the field of offline RL [40], where RL methods are used to learn from datasets of previously collected data. This data can include demonstrations, as well as other suboptimal trajectories. Such offline RL methods are particularly appealing for hybrid IL+RL methods, since they do not require any explicit distinction between demonstration data and suboptimal experience.

In this paper, we build on these approaches to develop a scalable algorithm that combines IL and RL. Our goal is to smoothly transition between two sources of supervision: the demonstrated behaviors, provided as prior data, and RL exploration, collected autonomously. Like many offline RL algorithms, our approach is based on Q-learning, which extracts the policy from a Q-function that is trained to satisfy the Bellman equation: $Q^\pi(s_t, a_t) = R(s_t, a_t) + \gamma \max_a[Q^\pi(s_{t+1}, a)]$.

To develop an effective IL-RL algorithm, we start by analyzing various design decisions in two recently proposed offline RL methods: QT-Opt [3] and Advantage-Weighted Regression (AWR) [1], with its recent value-based instantiation AWAC [2]. QT-Opt is a distributed Q-learning framework that enables learning Q-functions in continuous action spaces by maximizing the Q-function using the cross-entropy method (CEM) [41], without an explicit actor model. However, initializing a Q-function from demonstration data is particularly challenging as it often results in over-optimistic Q-values on unseen state-action pairs [42, 24, 22]. It has been shown that by introducing an explicit actor, one can mitigate this issue by limiting the actor to select actions that are within the (conditional) distribution of actions seen in the data [24, 1]. This property is also leveraged by AWAC [2], where the actor is initialized from demonstration data, and then further fine-tuned with RL rewards. The AWAC algorithm uses the following update to the actor:

$$\theta^\star \leftarrow \arg\max_\theta E_{s \sim \mathcal{D}, a \sim \pi_\beta(a|s)} \left[ \frac{1}{Z(s)} \exp\left(\frac{1}{\lambda} A^{\pi_\theta}(s, a)\right) \log \pi_\theta(a|s) \right], \tag{1}$$

where $A(s, a) = Q(s, a) - E_{\pi(a|s)}[Q(s, a)]$ is the advantage function, $Z(s)$ is the normalizing partition function, $\lambda$ is a Lagrange multiplier, $\pi_\theta$ is the learned policy and $\pi_\beta$ is the behavior policy, which represents the (unknown) distribution that produces the dataset. This corresponds to weighted maximum likelihood training, using samples from $\pi_\beta(a|s)$ (obtained from the dataset).

In this paper, we analyze various design decisions in QT-Opt and AWAC, point out their shortcomings in the IL+RL setting, and use these insights to construct an effective hybrid IL+RL method that we call AW-Opt.

# 4 Learning via Imitation and Reinforcement with AW-Opt

In order to derive AW-Opt, we first study the performance of QT-Opt [3] and AWAC [2] in the IL+RL setting, using a complex robotic manipulation task with high-resolution image observations as our working example. This setting requires algorithms that are more scalable than those typically employed in standard RL benchmarks. We will show that neither QT-Opt nor AWAC effectively handle this setting. We then progressively construct a new algorithm with elements of both approaches, which we call AW-Opt, analyzing each design decision in turn. Although each individual design decision in AW-Opt may appear relatively simple, we will see that the specific combination of these design decisions outperforms *both* QT-Opt and AWAC by a very large margin, resulting in successful learning for tasks where the two prior methods are unable to make meaningful progress.

## 4.1 Example Task and Baseline Performance

As a working example, we use a vision-based trash sorting task, where the robot must pick up a specific type of trash object (e.g., a compostable objects) out of cluttered bins containing diverse object types, including recyclables, landfill objects, etc., with only sparse rewards indicating success once the robot has lifted the correct object. We describe this task in detail

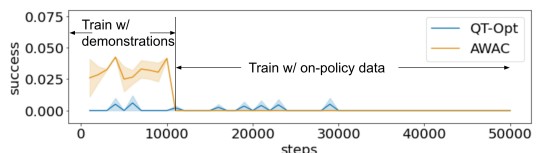

**Figure 2:** Baseline comparison for the example task. Neither QT-Opt [3], nor AWAC [2] can solve the task.

in Section 5. We will use this environment to illustrate the importance of each component of our method, as we introduce them in this section. Then, in Section 5, we will evaluate our complete AW-Opt algorithm on a wider range of robotic environments, including two real-world robotic systems and several high-fidelity simulation environments. Note that unless otherwise specified, all the plots in this paper is conducted with 3 experiments each. Each data point is an averaged success rate evaluated with 700 episodes. The shaded region in the plot depicts the 90% confidence interval with the line being the mean.

We begin our investigation with the two prior methods: QT-Opt [3], which can scale well to high-resolution images and diverse tasks, and AWAC [2], which is designed to integrate demonstrations and then fine-tune online. Both algorithms are provided with 300 demonstrations for offline pre-training, and then switch to on-policy collection after 10k gradient steps to collect and train with 200k additional on-policy episodes. In Fig. 2, we see that basic QT-Opt fails to make progress on both the pre-training and online data. Q-learning methods are typically ineffective when provided with only successful trials [40], since without both "positive" successful trajectories and "negative" failed trajectories, they cannot determine what *not* to do. Then, during the online phase, any RL method that is not pretrained properly is faced with a difficult exploration problem, and cannot make progress. QT-Opt is one of those RL algorithms. AWAC [2] does attain a non-zero success rate from the demonstrations, but performance is still poor, and it does not scale successfully to the high-dimensional and complex task during online fine-tuning. Next, we will introduce a series of modifications to AWAC that bring it closer to QT-Opt, while retaining the ability to utilize demonstrations, culminating in our full AW-Opt algorithm.

## 4.2 Positive Sample Filtering

One possible explanation for the poor performance of AWAC in Fig. 2 is that, with the relatively low success rate (around 4%) after pretraining, the addition of large amounts of additional failed episodes during online exploration drowns out the initial successful demonstrations and, before the algorithm can learn an effective Q-function to weight the actor updates, results in updates that remove the promising pre-training initialization. To address this issue, we introduce two modifications: **(a)** We use a

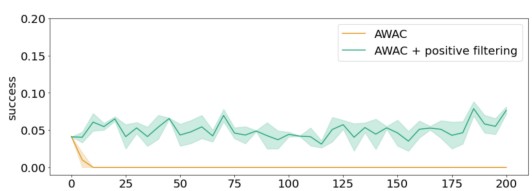

**Figure 3:** Positive filtering prevents AWAC performance from collapsing during the offline phase, as seen in Fig. 2. However, AWAC + positive filtering does not improve during the on-policy phase.

prioritized buffer *for the critic*, where 50% of the data in each batch comes from successful (positive) episodes that end with a reward of 1. Note that for tasks with non-binary rewards, we would need to introduce an additional hyperparameter for the "success" threshold, which we leave for future work. **(b)** We use positive filtering *for the actor*, applying the AWAC actor update in Equation 1 only on samples that pass the success filter (i.e., receive a reward of 1). While modification

**a)** ensures that the critic's training batch is balanced in terms of positive and negative samples, the modification in **b)** plays a subtle but very important role for the actor. By filtering the data that is presented to the actor, this modification ensures that the actor's performance should not drop below behavioral cloning performance, making it independent of the potentially inaccurate critic during the early stages of training.

After applying the above adjustments, AWAC performance no longer collapses after switching to online fine-tuning, as shown in Fig. 3, but still fails to improve significantly, with a final success rate of 5%. We abbreviate this intermediate method as AWAC_P (AWAC + **P**ositive filtering). To improve it further, we will discuss how to improve exploration performance in the next subsection.

### 4.3 Hybrid Actor-Critic Exploration

As a Q-learning method, QT-Opt does not have an explicit actor. Since the task involves continuous actions, QT-Opt uses the cross-entropy method (CEM) [41] to optimize the action using its critic, which can be viewed as an implicit policy (CEM policy). Unlike QT-Opt, AWAC is an actor-critic algorithm, which means it has access to both an explicit actor network and a critic network. Exploration is performed by sampling actions from this actor. However, this can present an obstacle to learning in complex

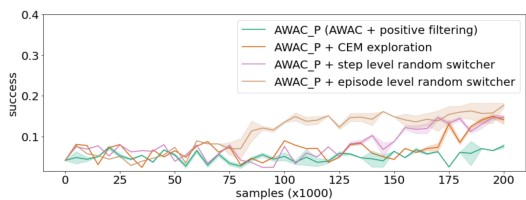

**Figure 4:** Comparisons of different hybrid actor-critic exploration strategies during the online fine-tuning phase. Using both the actor and critic for exploration significantly improves performance.

domains. First, the commonly used unimodal actor distribution may not capture the multimodal intricacies of the true Q-function, especially early on in training when the optimal action has not been determined yet. Second, the bulk of the learning must be done by the critic, since the actor doesn't reason about long-horizon rewards. Hence, the critic must first learn which actions are good, and then the actor must distill this knowledge into an action distribution, which can introduce delays.

Here, we aim to address these issues by utilizing *both* the actor and the critic for exploration. Although we are building on AWAC, we can still use the implicit CEM policy from QT-Opt to select actions via the critic, bypassing the actor. We therefore compare five exploration strategies: **(a)** actor-only exploration **(b)** implicit critic policy (as in QT-Opt, where CEM is used to pick the best action according to the critic); **(c)** episode-level random switcher, which randomly picks a policy at the beginning of each episode (80% critic policy, 20% actor); and **(d)** step-level random switcher, which randomly picks a policy at each time step (80% critic policy, 20% actor).

Fig. 4 shows the results of this comparison. The episode level random switcher outperforms the other candidates, achieving more than 10% success within 80k samples. Adopting the episode-level random switcher, we call this intermediate algorithm AWAC_P_ELRS. Given the observed improvement in the early stage of exploration, we shift our focus to analyzing whether value propagation in the critic is sufficiently able to utilize the data during online fine-tuning.

### 4.4 Action Selection in the Bellman Update

In the previous section, we showed how using the critic directly for exploration, rather than the actor, can lead to significantly better performance. We speculate that this is due to the fact that such an exploration strategy provides a degree of "lookahead": the actor can only learn to take good actions after the Q-function has learned to assign such actions high values. This means that by taking actions according to the Q-function during exploration, we can "get out ahead" of the actor and explore more efficiently. Can we apply a similar logic during training, and construct target Q-values with some amount of CEM optimization over actions,

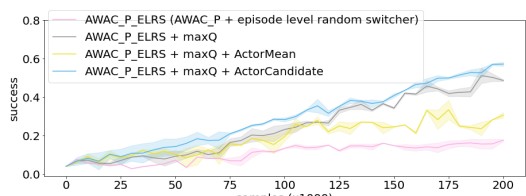

**Figure 5:** Comparison of different target value calculations. Maximizing the critic values generally performs significantly better than the standard AWAC target value, with the best version being maxQ + ActorCandidate which uses the actor as an additional action candidate during the CEM optimization.

rather than simply sampling actions from the actor?

Based on this intuition, we hypothesize that, in much the same way that direct maximization over the Q-function provided better exploration, improving over the actor's action in the target value

calculation will also lead to faster learning. While previous offline RL techniques [24, 2] make sure to carefully calculate the target value to avoid a potential use of out of distribution actions, we expect this to be less critical for our method, since the positive-filtering strategy in Section 4.2 already ensures that the actor is only trained on successful actions, significantly limiting the harm caused by an inaccurate Q-function. To evaluate this idea, we test several ways of selecting actions in the Bellman backup.

In addition to the standard AWAC target value, which is taken as an expectation under the actor, we test: **(a) maxQ** uses the same target value calculation as QT-Opt, optimizing the action in the target value calculation with CEM ("maxQ"); **(b) maxQ + ActorMean** combines the actor with CEM by using the actor-predicted action as the initial mean for CEM; **(c) maxQ + ActorCandidate** combines the actor with CEM by using the actor-predicted action as an additional candidate in each round of CEM. These target value calculations, particularly **(a)**, bring the method significantly closer to QT-Opt. In fact, **(a)** can be equivalently seen as a QT-Opt Q-function learning method, augmented with an additional positive-filtered actor that is used for exploration. However, as we see in Fig. 5, the performance of this approach is significantly better than both the AWAC-style Bellman backup and the standard QT-Opt method, neither of which learn the task successfully. The

---

**Algorithm 1** AW-Opt

1: Dataset $\mathcal{D} = (s_t, a_t, r(s_t, a_t), s_{t+1})$
2: Initialize buffer $\beta = \mathcal{D}$
3: Initialize parameter vectors $\phi, \theta, \bar{\theta}$
4: **for** each iteration `iter` **do**
5:    **if** `iter` $>= T$ **then**
6:      **for** each environment step **do**
7:        $a_t \sim \pi_\phi(a_t|s_t))$ or $a_t \sim \pi_{CEM}(a_t|s_t))$
8:        $s_{t+1} \sim p(s_{t+1}|s_t, a_t)$
9:        $\beta \leftarrow \beta \cup (s_t, a_t, r(s_t, a_t), s_{t+1})$
10:      Sample batch $(s_t, a_t, r(s_t, a_t), s_{t+1}) \sim \beta$
11:    **for** each gradient step **do**
12:      $a^* = \arg\max_a(Q_\theta(s_{t+1}, a))$
13:      $\theta_i \leftarrow \theta_i - \lambda_Q \nabla L_Q(Q_{\theta_i}(s_t, a_t), r(s_t, a_t) + \gamma Q_{\theta_i}(s_{t+1}, a^*)$
14:      $\bar{\theta} = \tau\theta + (1 - \tau)\bar{\theta}$
15:      Adv = $Q_{\bar{\theta}}(s_t, a_t) - E_{s_t \sim \beta, a_t \sim A_\phi(s_t)}[Q_{\bar{\theta}}(s_t, a_t)]$
16:      $\phi_i \leftarrow \phi_i - \lambda_A \exp(\text{Adv}) \nabla L_A(a_t, A_{\phi_i}(s_t)) \cdot \mathbb{1}_{\text{success}}$

---

"maxQ" strategy leads to more than three-fold increase in performance after 200k samples over the previous best method. In addition, we notice that **(b)** is worse than **(a)**. Our hypothesis for why "maxQ + ActorMean" is worse is the following: in actor-critic methods, the actor always lags behind the critic. Therefore, when the actor is not good enough, it may shift the initial distribution and lead the CEM in the wrong direction.

Fig. 5 demonstrates the comparison of these techniques. AWAC_P_ELRS + maxQ is significantly better than AWAC_P_ELRS. Two variants: AWAC_P_ELRS + maxQ + ActorMean (using actor-predicted action as the initial CEM mean) and AWAC_P_ELRS + maxQ + ActorCandidate (using actor-predicted action as an additional CEM candidate to compare with the best action selected by CEM) also improve over AWAC_P_ELRS. As we notice that AWAC_P_ELRS + maxQ + ActorCandidate achieves the best performance, we turn AWAC_P_ELRS + maxQ + ActorCandidate into our final AW-Opt method, which is shown in Algorithm 1.

## 4.5 Ablations

After the step-by-step design of the AW-Opt algorithm, we run an additional ablation study by removing one feature at a time to further identify the importance of each of the design choices we have made. We compare the following four cases: **(a)** AW-Opt, **(b)** AW-Opt without positive-filtering, **(c)** AW-Opt without ActorCandidate, **(d)** AW-Opt without hybrid actor-critic exploration.

As shown in Fig. 6, each of the introduced components plays a significant role in providing a

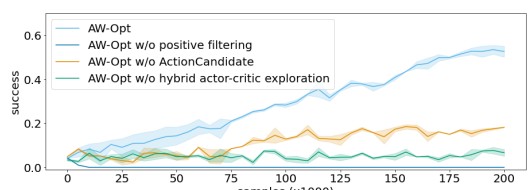

**Figure 6:** Ablations of each individual component of AW-Opt. Note that every ablation significantly degrades performance, indicating the importance of each of the components.

boost in performance. Among the proposed AW-Opt design decisions, positive filtering is particularly crucial for achieving good results in the online phase. Without applying positive filtering, the performance of AW-Opt collapses to 0% once on-policy training starts, confirming our hypothesis that positive-filtering of the actor robustifies the actor updates against inaccurate Q-functions in the initial phases of training. However, each of the other components is also critical, with the best-performing ablation (the one that uses the standard AWAC target value calculation) still being about three times worse than full AW-Opt.

# 5 Experiments

In our experimental evaluation, we study the following questions: (1) Can AW-Opt learn from both expert demonstrations and suboptimal off-policy data? (2) Is AW-Opt a viable RL fine-tuning algorithm that continues to improve after pre-training on offline data in diverse real-world and simulated settings? (3) Is AW-Opt competitive on metrics important for robot learning, such as real-world performance and sample efficiency? (4) Are the previously mentioned design choices critical for the success of the method on a wide range of tasks?

## 5.1 Experimental Setup

We evaluate our method on simulated and real robotic manipulation tasks (Fig. 1 and Fig. 7). We use two real robotic systems with parallel-jaw grippers and over-the-shoulder cameras: Kuka IIWA robots, and a proprietary 7-DoF arm that we call "Pica" [1]. We use two simulated tasks and three real tasks in our experiments:

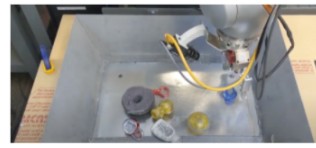

*Task 1:* **(Sim) Indiscriminate Grasping** with Kuka (Fig. 1 (a)). Six random procedurally-generated objects are placed in a bin. If the robot lifts any of the objects, the episode is successful. We utilize around 1000 off-policy simulation trials for this task.

*Task 2:* **(Sim) Green Bowl Grasping** with Pica (Fig. 1 (b)). A green bowl is randomly placed on a table among other objects. If the robot lifts the bowl, the episode is considered a success. We utilize 120 teleoperated simulated demonstrations for this task.

*Task 3:* **(Real) Indiscriminate Grasping** with Kuka (Fig. 1 (c)). This is the real-world version of Task 1. We follow the setup proposed in [3], and utilize off-policy data for this task for pre-training.

*Task 4:* **(Real) Grasping from the Middle Bin** with Pica (Fig. 1 (d)). Several types of trash are placed randomly in three bins. The episode is successful if the robot lifts any object from the middle bin. We use 1000 demonstrations for this task.

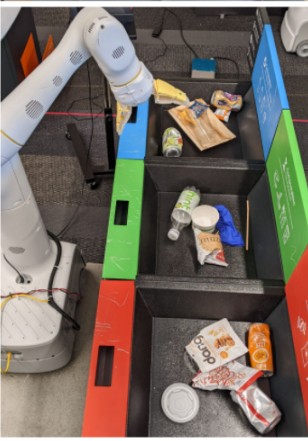

**Figure 7:** Real robot setup.

*Task 5:* **(Real) Grasping Compostables from the Middle Bin** with Pica (Fig. 1 (d)). This is a more difficult version of Task 4. While setup is equivalent, an episode is successful if a compostable object was lifted. We utilize 300 demonstrations for this task.

In the pretraining phase, we utilize the prior data described above and train the network (the same network as QT-Opt [3]) for 5,000 to 10,000 gradient steps. Each algorithm then collects 20,000 to 200,000 episodes of on-policy data during further training. For Task 4 and Task 5, fine-tuning is done in simulation, using RL-CycleGAN [43] to produce realistic images. This allows us to compare fine-tuning performance between QT-Opt, AWAC, and AW-Opt on these tasks, using simulated evaluations. Separately, we conduct real-world evaluations, evaluating the offline pretraining stage and sim-to-real transfer of fine-tuned policies, since running each of the three algorithms online in the real world would be too time-consuming. This evaluation protocol provides evidence to support our central claims: the simulated experiments confirms that our method can effectively finetune from online data, while the real-world evaluation confirms that our approach can handle offline pre-training on real data effectively.

## 5.2 Learning from Demonstrations and Suboptimal Off-Policy Data

To address the question of whether AW-Opt can learn using different types of prior data, we prepare datasets from different sources: simulated off-policy data (Task 1), simulated demonstrations (Task 2) and real demonstrations (Task 4 and Task 5). We compare the learning performance of QT-Opt, AWAC, and AW-Opt using these 3 different data sources in Fig. 8 (left panels of each subplot). Both AWAC and AW-Opt can learn from both demonstrations and off-policy data, while QT-Opt cannot, since Q-learning requires both successful and failed trials to determine which actions are better than others. Depending on the difficulty of the task, the success rates of the policies learned by AWAC and AW-Opt vary from 20% to 60% with the exception of the difficult Task 5, which requires on-policy finetuning.

---

[1]Pica is a pseudonym for this paper.

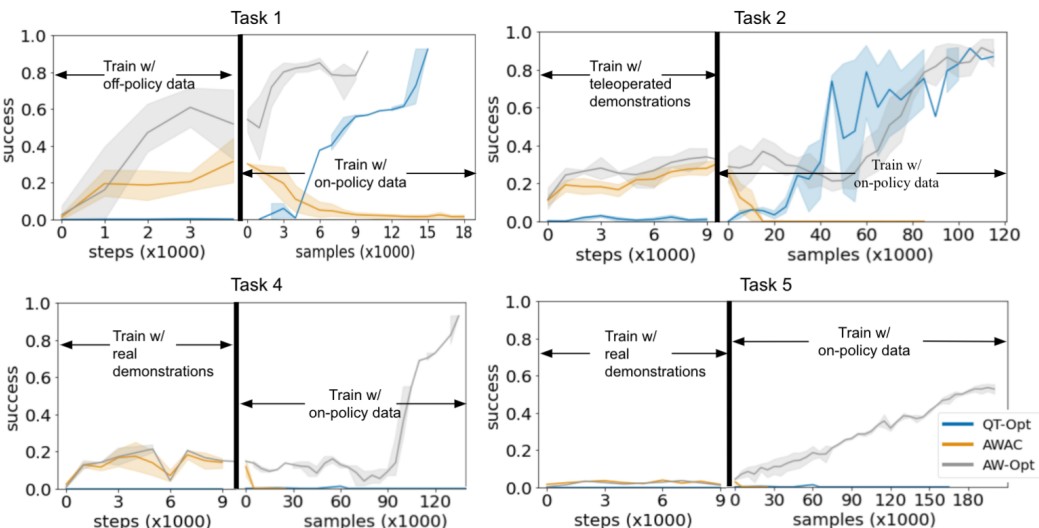

**Figure 8:** Offline pretraining followed by online finetuning on each of our simulated evaluation tasks. Each plot shows the offline phase (left of the vertical line), followed by an online phase where each method uses its respective exploration strategy to collect more data. In most tasks, AW-Opt and AWAC learn a moderate level of success from offline data, while QT-Opt struggles due to the need for negatives. During online finetuning, QT-Opt generally performs better than AWAC, but AW-Opt outperforms both prior methods, both learning faster (e.g., for Task 1) and reaching a significantly higher final performance (Take 4 and Task 5).

## 5.3 Online Fine-tuning

As shown in Fig. 8 (right panels of each subplot), during online fine-tuning, QT-Opt performs significantly better than AWAC, while the latter struggles to effectively utilize online rollouts. However, with the modifications in AW-Opt, our method is able to attain the best of both worlds, fine-tuning effectively during the online phase, while utilizing the offline data to bootstrap this fine-tuning process. While AW-Opt is more data-efficient than QT-Opt on Task 1 and Task 2, this difference is particularly pronounced on Task 4 and Task 5, where the inability of QT-Opt to utilize the prior data significantly hampers its exploration.

## 5.4 Real-World Robotic Manipulation Results

We evaluate the performance of policies trained with QT-Opt, AWAC, and AW-Opt after offline pre-training for Task 3 and after finetuning for Task 5. The fine-tuning stage is performed in simulation, as discussed in Section 5.1. The goal of these experi-

**Table 1:** Task 3 and Task 5 real robot results.

|        | Task 3 positives only | Task 3 positives + negatives | Task 5 finetune |
|--------|------------------------|-------------------------------|-----------------|
| QT-Opt | 0%                     | 61.23%                        | 0%              |
| AWAC   | 44.21%                 | 0%                            | 0%              |
| AW-Opt | 52.53%                 | 57.84%                        | 48.89%          |

ments is to validate that policies learned with AW-Opt can indeed handle offline demonstration data and generalize to real-world settings. The size and type of the datasets are described in Appendix. For Task 3, we first train AWAC, QT-Opt, and AW-Opt with only positive data and then train with both positive and negative data. We evaluate each policy with 612 grasps, as shown in Table 1. When learning from the combined positive and negative data, AWAC performance drops to 0%. When QT-Opt learns from only positive data, it also fails to learn the task. Only AW-Opt is able to learn from both positive-only and mixed positive-negative data. For Task 5, we evaluate each policy with 180 grasps, as shown in Table 1. Only AW-Opt learns this task successfully, achieving a 48.89% success rate. These results support the conclusions drawn from our simulated experiments.

## 6 Conclusion

We presented AW-Opt, a scalable robotic RL algorithm that incorporates offline data and performs online finetuning. We show that each design decision in AW-Opt leads to significant gains in performance, and the final method can learn complex image-based tasks starting with either demonstrations or suboptimal offline data, even in settings where prior methods, such as QT-Opt or AWAC, either fail to leverage prior data or fail to make progress during finetuning. Our evaluation does have several limitations. First, we only evaluate offline training in the real world, studying finetuning under simulated settings where extensive comparisons are feasible. Our experiments are also all concerned with sparse, binary reward tasks. Such tasks are common in robotics, and AW-Opt could be extended to arbitrary rewards via a threshold for positive filtering, a promising direction for future work. Nonetheless, our results suggest that AW-Opt can be a powerful tool for scaling up IL+RL methods. Aside from motivating each design decision, we also hope that the detailed evaluations of each component will aid the design of even more effective and scalable RL methods in the future.

**Acknowledgments**

We would like to give special thanks to Dao Tran, Conrad Villondo, Clayton Tan for collecting demonstration data as well as Julian Ibarz and Kanishka Rao for valuable discussions.

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
