# OpenReview forum: "AW-Opt: Learning Robotic Skills with Imitation andReinforcement at Scale"
_robot-learning.org/CoRL/2021/Conference — CoRL2021 Poster_

### Official Review · Reviewer_qeFD · 2021-07-25

**Originality:** Fair
**Technical Quality:** Good
**Clarity Of Presentation:** Good
**Impact:** 2

**Recommendation:**

Weak Accept: I recommend accepting the paper, but will not argue for my recommendation if the majority of other reviewers have a different opinion.

**Summary:**

This paper, on the one hand, investigates experimentally why two existing RL methods have difficulties solving a bin picking task with a sparse reward for pick-up success.
On the other hand, the authors compare performance improvements by introducing small changes in three different aspects of the RL problem:
1) Data balancing for training the Q function and the policy.
2) Exploration with a learned actor policy vs optimizing the learned Q function.
3) Different ways of calculating the Bellman update.

In the experiments, the influence of these changes both in terms of asymptotic performance and the capability of the system to include demonstration data is investigated.


**Issues:**

- "Then, during the online phase, QT-Opt is faced with a difficult exploration problem". Why QT-opt particularly?
- The fact that the episode-level random switcher is the best in Fig. 4 seems to be caused by the fact that it introduces the most useful randomness for exploration compared to CEM exploration only? How has the 80% 20% ratio been chosen?
- In Fig. 5 why is the grey curve better than the yellow one given the fact that the blue one outperforms all? Maybe a clarification how actor candidate differs from actor mean would be useful. (particularly I don't full get what it is meant with "maxQ + ActorCandidate combines the actor with CEM by using the actor-predicted action as an additional candidate in each round of CEM.").
- Why not include non successful demonstrations for QT-opt (as done in Tab. 1)? It seems arbitrary to not do that for most of the experiments and indeed in sec. 5.4 QT-opt performs best on task 3.
- Line 13 in Algo 1 does not make sense (typo, I think).

**Reviewer Expertise:**

Good: General knowledge of the area

**Strengths And Weaknesses:**

Strengths:
- I appreciate the research question of figuring out why existing methods have difficulties in incorporating demonstrations for later fine tuning.
- Interesting ablation studies/comparisons.
- Good video!

Weaknesses:
- No real methodological advancement. Proposed approach consists of only small tweaks to existing methods.
- Results don't allow a clear conclusion (QT-opt can also incorporate negative demonstrations).
- The choice of QT-opt and AWAC seem rather arbitrary.

**Summary Of Recommendation:**

I am not sure what to think about this paper.
The gained insights are not really surprising.
It is well-known that RL is difficult due to the exploration problem of collecting meaningful data, the credit assignment problem and the learning problem itself (whether it is Q functions or policies, even given optimal (but imbalanced) data).
What I have learned from the paper is that small modifications can make a large difference without changing the underlying learning methods significantly. But this also implies that the problem is not yet well understood yet and I am unsure if the findings of the paper are applicable to other domains/tasks as well.
Nevertheless, the approach of investigating the data balance problem, exploration and Bellman update is interesting and relevant.
I would recommend ICRA instead of CoRL.

---

> ### Author Response · Authors · 2021-08-26
> **quFD response**
>
> Thank you for the detailed feedback. We added additional experiments to study how QT-Opt can incorporate negatives, and we address each of the concerns raised in your review below.
>
> **Methodological advancement:**
> It’s true that the technical decisions made in our method consist of a set of seemingly small changes to existing methods. But that’s the whole point of the paper: while these decisions may seem minor, they result in very large changes in practical performance (see, e.g., Fig 1(f) for a concise summary). We show that a detailed point-by-point evaluation of each of these decisions allows us to drastically improve the performance of IL+RL methods on challenging and realistic robotic tasks, on which prior methods either fail to learn with RL (AWAC) or fail to utilize demonstrations (QT-Opt). The whole point of the paper is that all of these details matter a lot, and besides the specific choices that we show work well in our experiments (which are evaluated thoroughly), we hope that our paper will also serve as a case study for other robotics researchers as an example of how to systematically design the crucial details in RL methods that matter so much in practice. As acknowledged by the other reviewers, such a systematic study that leads to a method with good empirical performance has a lot of value for the robotics community.
>
> QT-Opt can also incorporate negatives:
> We believe there is a subtle misunderstanding here, but we added an additional experiment to study this point, which we discuss below.
>
> Task 3 data was collected with a scripted policy that is highly randomized and frequently fails. Such data is very well-suited for naive offline Q-learning, because it has good coverage (see, e.g., discussion in [Kumar et al. “Stabilizing Off-Policy Q-Learning…”]). Indeed, this is why it was used in the original QT-Opt paper – this data distribution was designed with QT-Opt in mind! This is the second column Tbl 1. To “simulate” what would happen if we instead had demo data, we excluded the failures. This is the second column in Tbl 2. However, for most real demonstration datasets (i.e., demos from humans rather than a scripted policy), such “negatives” just don’t exist. It’s not enough to just get data that fails, this data must also have good coverage.
>
> To study this, we conducted a study on Task 4 where we added random failure episodes, which is a realistic source of negatives in such cases. In this case, we find that QT-Opt does not improve over its performance with only positives, as can be seen in the updated manuscript Appendix A.3.
>
> In summary, our evaluation is fair and provides realistic data to all methods in the form of demonstrations, but the somewhat contrived evaluation setup in Task 3 allows us to get “ideal” negatives that allow QT-Opt to work. This is not possible in most tasks, and naively provided negatives are not sufficient for QT-Opt to work well, as demonstrated in the new experiment above. We would be happy to evaluate other ways to obtain negatives, but we are not aware of any such methodology in the literature.
>
> **Choice of QT-Opt and AWAC:**
> We chose AWAC because it was previously studied (in the original AWAC paper) as a method that enables pretraining offline from demonstrations followed by online finetuning. We chose QT-Opt because it achieves good results on some of the tasks we studied with online training, making it a good choice. We did try a number of other algorithms (SAC, DDPG, TD3), but were unable to obtain good results with these methods on these tasks. It may be that a similar sequence of improvement steps as we propose for AW-Opt could improve these other methods as well however, and we will add discussion of this point to the paper. However, it is not clear what we should have done better in this case – is there a specific modification that you believe would better address this issue?
>
> **ICRA vs CoRL:**
> While we appreciate the reviewer’s recommendation about publication venue, we believe CoRL is the most suitable venue for our work, because our work addresses robotic learning specifically, and we believe that robotic learning practitioners would find the detailed experiments in our paper to be valuable guidance both for implementing IL+RL methods, and as a case study for how to develop an effective RL algorithm. That said, we are glad to see that the reviewer believes that our work is above the bar for publication, and the main question is the publication venue.

---

> > ### Author Response · Authors · 2021-08-26
> > **Other details**
> >
> > **> "Then, during the online phase, QT-Opt is faced with a difficult exploration problem". Why QT-opt particularly?**
> >
> > We will adjust the phrasing here – all methods face a difficult exploration problem if they are not properly pretrained, there is nothing special about QT-Opt here. The point we are trying to make here is that any method that fails to pretrain from the demos will struggle to finetune online, because its poor initialization does not enable it to see any positive reward on sparse reward tasks (such as the ones we study).
> >
> > **> The fact that the episode-level random switcher is the best in Fig. 4 seems to be caused by the fact that it introduces the most useful randomness for exploration compared to CEM exploration only? How has the 80% 20% ratio been chosen?**
> >
> > Thank you for pointing this out, we added an additional experiment to study this, which we describe below. Initially, the actor can learn from positive data and produce successful episodes while the critic cannot. Therefore, it can generate more positive data at the initial phase, making the critic (CEM) much easier to learn. Once the critic (CEM) learns faster and better, the actor can further benefit from it.
> >
> > The 80/20 ratio was chosen arbitrarily, and we did not have a chance to try any other ratio at the time of submission. However, we have now added additional experiments to address this, with results in the updated manuscript Appendix A.2.
> >
> > **> In Fig. 5 why is the grey curve better than the yellow one given the fact that the blue one outperforms all? Maybe a clarification how actor candidate differs from actor mean would be useful. (particularly I don't full get what it is meant with "maxQ + ActorCandidate combines the actor with CEM by using the actor-predicted action as an additional candidate in each round of CEM.").**
> >
> > We will clarify this in the paper. The CEM process is the following.
> > 1. Initialize a Gaussian distribution with some mean and variance.
> > 2. Draw samples from the Gaussian distribution
> > 3. Compute Q values for these samples
> > 4. Choose the top x% of the samples
> > 5. Compute mean and var for the elites
> > 6. If iter == max_iter Go to #7, else Go to #1
> > 7. Select the action w/ highest Q value as the best action
> > The "maxQ + ActorCandidate" method includes the action predicted by the actor as an additional sample in step 2, such that the actor’s action may be included in the set of elites. The "maxQ + ActorMean" method uses the mean and variance of the actor as the initial distribution in step 1 (else we use an uninformative distribution).
> >
> > Our hypothesis for why "maxQ + ActorMean" is worse is the following: in actor-critic methods, the actor always lags behind the critic. Therefore, when the actor is not good enough, it may shift the initial distribution and lead the CEM in the wrong direction (i.e., a poor local optimum).
> >
> > **> Why not include non successful demonstrations for QT-opt (as done in Tab. 1)? It seems arbitrary to not do that for most of the experiments and indeed in sec. 5.4 QT-opt performs best on task 3.**
> >
> > See discussion above. As shown in our new experiment, simply adding failed episodes is not enough. The Task 3 dataset is just unusual in that it was collected with a high-coverage scripted policy that allows this to work, which is not the case for real demonstration datasets.
> >
> > **> Line 13 in Algo 1 does not make sense (typo, I think).**
> >
> > Thanks. We fixed Line 13 in Algo 1.

---

> ### Author Response · Authors · 2021-08-27
> **Does the rebuttal address your concerns?**
>
> Dear Reviewer,
>
> We sincerely appreciate your time for the review, and we really hope to have a further discussion.
>
> Could you kindly check our responses and let us know whether our rebuttal addresses your concerns?
>
> Thanks a lot,
>
> Authors

---

> > ### Comment · Reviewer_qeFD · 2021-09-02
> > **Re: Does the rebuttal address your concerns?**
> >
> > Thank you for your answers, which address most of my concerns.
> >
> > I have updated my score to weak accept.
> >
> > I would encourage the authors to check Algorithm 1 again for typos (I think there are still bracket typos at some places).

---

### Official Review · Reviewer_qTLQ · 2021-07-26

**Originality:** Good
**Technical Quality:** Very Good
**Clarity Of Presentation:** Excellent
**Impact:** 4

**Recommendation:**

Strong Accept: I recommend accepting the paper and will argue for my recommendation even if other reviewers hold a different opinion.

**Summary:**

This paper proposes a hybrid method AW-Opt, built on two previous methods AWAC and Qt-Opt, for scalable robotic manipulation with imitation learning and reinforcement learning.
The resulting method is driven by empirically analyzing the design choices in both baseline methods and developing modifications that address the issues in the original components. Specifically, the proposed changes are balanced data for training both the critic and actor, improved exploration by switching between the actor and CEM policy, and bellman update target using maximum Q value optimized by CEM with "elites" from the actor.

With extensive experiments in simulation and real-world, the paper demonstrates that the proposed method significantly improves over the baselines AWAC and Qt-Opt in challenging tasks where offline learning and online finetuning are needed for achieving high performance.

**Issues:**

Please refer to the items discussed in the weaknesses/suggestions section. My main concern is in the missing details of the learning curves.

**Reviewer Expertise:**

Very good: Comprehensive knowledge of the area

**Strengths And Weaknesses:**

Strengths
1. The presentation of the paper is clear
2. The modifications proposed in this paper are well motivated and they are effective as shown by extensive experiments
3. The complete algorithm archives strong empirical performance

Weaknesses/Suggestions
1. It missed some details that are important for interpreting the assessing the experimental results:

  a. Most figures showed the success rate but did not give details on how many trials the rate was based on.
  b. All learning curves were shown without mentioning what the line and shaded region represent. Is it the mean and standard deviation? If so, how many runs/random seeds were they based on?

2. Can the authors please clarify the following design choices in the paper?
  a. The “switcher” exploration in Section 4.3 used 80% critic and 20% actor. I wonder if this choice is backed up by some experiment. How sensitive is the result to this choice of ratio?
  b. The offline datasets for the five tasks contain different positive/negative ratios. I wonder if this is a somewhat arbitrary choice (in terms of that the extreme case of 100% negative is bad but other ratios are fine). How sensitive is the result to this choice of ratio when constructing the dataset?
  c. The loss function shown in the appendix has weights for subactions. It would be great if the values of the weights are given.

3. The legend in Figure 6 does not quite match the text description. “AW-Opt - MaxQ” in the figure becomes “AW-Opt without ActorCandidate” in the text. It seems to me that they are not the same thing: we recover the AWAC target value without the MaxQ but “without ActorCandidate” can be interpreted as AW-Opt + MaxQ. It confuses me as to which one the paper meant to refer to.


**Summary Of Recommendation:**

I base my recommendation mainly on the strong experimental results and the potential impact of the proposed ideas and insights. In the robotics domain, simple but effective methods often have great impacts. This paper falls into this category.

---

> ### Author Response · Authors · 2021-08-26
> **qTLQ response**
>
> Thank you for your questions and feedback. Please see our responses below:
>
> **Details on experimental results:**
> Each point in the figures corresponds to 700 trials and we ran three random seeds for each curve. The shaded region in the plot depicts the 90% confidence interval with the line being the mean.
>
> **Design choice clarification:**
> Yes, you are correct - the positive/negative ratio was picked arbitrarily based on the available data. We did not observe sensitivity of the algorithm to this ratio. To clarify the loss question, we added the values of the loss weights in the appendix.
>
> **Legend in Fig. 6**
> Thank you for pointing it out - we fixed the legend in the new version of the manuscript.

---

### Official Review · Reviewer_3em5 · 2021-08-09

**Originality:** Good
**Technical Quality:** Good
**Clarity Of Presentation:** Very Good
**Impact:** 4

**Recommendation:**

Weak Accept: I recommend accepting the paper, but will not argue for my recommendation if the majority of other reviewers have a different opinion.

**Summary:**

In this paper, the authors have tried to combine the advantage of imitation learning (IL) and reinforcement learning (RL) into a single framework. Mainly authors have focused on Advantage-Weighted Actor-Critic (AWAC), which learns good initial policy from demonstration, and QT-Opt, which can learn a task from scratch with a large amount of training data, to develop AW-Opt. To make AW-Opt fair well with IL plus RL, authors have modified data sampling, exploration strategy, and action selection in the Bellman equation. Authors have shown that these modifications helped robotic manipulators to solve 5 different robotic tasks, consist of both simulator and real-world, when provided with initial demonstration and then finetuned using RL.

**Issues:**

One of the major concerns I have is that in this paper authors are proposing a new algorithm for solving IL + RL problem, but has only shown result based out of the one vision-based robotic grasping task with some amount of variation in what needed to be grasped. However, it would be interesting to know how this algorithm works for some complex manipulation tasks like opening microwave, cabinet, door or push stacking objects.

**Reviewer Expertise:**

Good: General knowledge of the area

**Strengths And Weaknesses:**

Strengths
* Figured out the limitation and advantages of each technique and combined them well to figure out technique which works fairly well for IL and RL setup
* Provide ablation study to show the importance of each strategy
* Carried out experiment both in simulation and real world robotic platform.

Waekness
* It felt to me that method was overly optimized for the task in consideration. I would love to know how the method works for wide range of IL  + RL task, ranging from learning policy for solving video games to complex manipulation (like opening doors, cabinets, etc)
* No good explanation for choosing some of the hyper parameters, like 20-80 % selection in Hybrid Actor-Critic Exploration

**Summary Of Recommendation:**

I am providing the weak accept based out of the following +ve and -ve points

### +ve points
* The paper proposes new way of performing IL and RL based out of vision inputs in sample efficient manner.
* They have shown that the method they are proposing, is able to solve 3 robotic task in real world based out of demonstration collected for those tasks
* Detailed ablation study is provide which shows that each of the strategy is contributing significant amount to the overall success

### -ve points
* seems method is overly optimized for task in consideration
* explanation for some of the hyper parameters are missing

---

> ### Author Response · Authors · 2021-08-26
> **3em5 response**
>
> Thank you for your review and feedback. Please see our responses below:
>
> In regard to generality and other tasks: to address this point, we added an additional experiment on a LIDAR-based navigation task, which is not a manipulation domain and presents distinct challenges. On this task, AW-Opt achieves a similar improvement over AWAC and QT-Opt as it did on the manipulation tasks. The results can be found in the updated manuscript Appendix A.1.
>
> Besides this, while our evaluation studies many of the design decisions in AW-Opt in great detail, several details were not studied, such as the choice of 80/20 split for Q-function vs. actor exploration. We added additional experiments to address this, with results in the updated manuscript Appendix A.2.

---

### Author Response · Authors · 2021-09-02
**General Response**

Dear AC,

We believe we've fully addressed the issues raised by Reviewer quFD: we added an additional robotic navigation task that does not involve grasping to address questions about generality, we added experiments studying the 80/20 exploration split, and we clarified all of the other questions and issues. We would appreciate a response from Reviewer quFD as to whether this fully addresses their concerns.

---

### Meta-Review · Area_Chair_LG9Y · 2021-08-13

**Recommendation:** Accept (Poster)
**Confidence:** 5

**Metareview:**

Reviewers are mixed. On the one hand, all reviewers appreciate the study on understanding why two well-known existing algorithms struggle to incorporate demonstrations well, and on proposing specific improvements to these algorithms accordingly. On the other hand, two reviewers question whether the conclusions would apply to other tasks, beyond the grasping task studied in this paper, and Reviewer qeFD notes a number of other minor potential weaknesses. Authors should address these issues in a rebuttal.

-------

Following the reviews, the authors have provided further experiments and have addressed the issues raised by the reviewers. All three reviewers now recommend acceptance of the paper. For the new navigation experiments (Figure 10) in the final paper, I encourage the authors to perform these over more than one random seed, as has been done for the manipulation experiments.

---

> ### Author Response · Authors · 2021-08-26
> **Meta-reviewer response**
>
> Thank you for the feedback. We address the detailed questions from the reviewers in responses to each reviewer below, while here we respond to high-level points made in the meta-review.
>
> In regard to generality and other tasks: to address this point, we added an additional experiment on a LIDAR-based navigation task, which is not a manipulation domain and presents distinct challenges. On this task, AW-Opt achieves a similar improvement over AWAC and QT-Opt as it did on the manipulation tasks. The results can be found in the updated manuscript Appendix A.1. We hope that this addresses the generality concerns.
>
> We also note that in the original submission, we already evaluate on two simulated and two real-world domains (see task list in Sec 5.1). While all of the tasks involve grasping, it is tested with different robots, and with different semantic goals (e.g., grasping a specific object vs grasping anything). This is a significant breadth of evaluation.
>
> Besides this, while our evaluation studies many of the design decisions in AW-Opt in great detail, several details were not studied, such as the choice of 80/20 split for Q-function vs. actor exploration, as pointed out by the reviewers. We added additional experiments to address this, with results in the updated manuscript Appendix A.2. Other specific points are addressed in individual reviewer responses, and we would be happy to add other requested evaluations.

---

### Decision · Program_Chairs · 2021-09-13

**Decision:**

Accept (Poster)

**Comment:**

Reviewers are mixed. On the one hand, all reviewers appreciate the study on understanding why two well-known existing algorithms struggle to incorporate demonstrations well, and on proposing specific improvements to these algorithms accordingly. On the other hand, two reviewers question whether the conclusions would apply to other tasks, beyond the grasping task studied in this paper, and Reviewer qeFD notes a number of other minor potential weaknesses. Authors should address these issues in a rebuttal.

-------

Following the reviews, the authors have provided further experiments and have addressed the issues raised by the reviewers. All three reviewers now recommend acceptance of the paper. For the new navigation experiments (Figure 10) in the final paper, I encourage the authors to perform these over more than one random seed, as has been done for the manipulation experiments.